# Enhanced TfR1 Recognition of Myocardial Injury after Acute Myocardial Infarction with Cardiac Fibrosis via Pre-Degrading Excess Fibrotic Collagen

**DOI:** 10.3390/biology13040213

**Published:** 2024-03-25

**Authors:** Wenwen Yang, Yueqi Wang, Hongzheng Li, Feifei Liao, Yuxuan Peng, Aimei Lu, Ling Tan, Hua Qu, Linzi Long, Changgeng Fu

**Affiliations:** 1Graduate School, China Academy of Chinese Medical Sciences, Beijing 100091, China; 2CAS Key Laboratory of Molecular Imaging, Beijing Key Laboratory of Molecular Imaging, The State Key Laboratory of Management and Control for Complex Systems, Institute of Automation, Chinese Academy of Sciences, Beijing 100190, China

**Keywords:** myocardial infarction, human serum albumin, collagenase I, transferrin receptor 1, near-infrared fluorescence imaging

## Abstract

**Simple Summary:**

After myocardial infarction (MI), myocardial tissue undergoes a series of complex biological reactions, among which fibrosis is particularly critical. This process leads to troublesome collagen deposition, which not only affects the normal function of the myocardium but also poses a serious challenge to conventional myocardial injury imaging techniques. Especially in the metabolism of contrast media, the uneven distribution caused by fibrosis greatly reduces the accuracy of imaging. To address this problem, we propose an imaging method for myocardial fibrosis processing after myocardial infarction, aiming to improve the accuracy of myocardial injury imaging. Firstly, a combination of collagenase I and human serum albumin (HSA-C) was used to perform deep clearance of myocardial fibrotic collagen. Collagenase I is an enzyme that specifically acts on collagen fibers and can effectively degrade collagen fibers in the myocardium. However, human serum albumin serves as a carrier to help collagenase better distribute in myocardial tissue. After the successful clearance of fibrotic collagen from the myocardium, optical contrast agents targeting the transferrin receptor were used to precisely localize the damaged myocardium. The expression of the transferrin receptor significantly increased after myocardial injury, so this contrast agent can specifically identify the injured myocardium. This precise localization capability provides an important basis for subsequent imaging techniques. With the use of near-infrared fluorescence imaging, we can precisely track the location of myocardial injury. In the area of myocardial damage, the fluorescence signal is significantly enhanced due to the specific binding of the contrast agent, to achieve accurate tracking of the location of myocardial injury. This pretreatment strategy not only effectively eliminates the background interference caused by contrast agent deposition in fibrotic tissue, but also improves the targeting efficiency of contrast agents in cardiac lesions.

**Abstract:**

The fibrosis process after myocardial infarction (MI) results in a decline in cardiac function due to fibrotic collagen deposition and contrast agents’ metabolic disorders, posing a significant challenge to conventional imaging strategies in making heart damage clear in the fibrosis microenvironment. To address this issue, we developed an imaging strategy. Specifically, we pretreated myocardial fibrotic collagen with collagenase I combined with human serum albumin (HSA-C) and subsequently visualized the site of cardiac injury by near-infrared (NIR) fluorescence imaging using an optical contrast agent (CI, CRT-indocyanine green) targeting transferrin receptor 1 peptides (CRT). The key point of this strategy is that pretreatment with HSA-C can reduce background signal interference in the fibrotic tissue while enhancing CI uptake at the heart lesion site, making the boundary between the injured heart tissue and the normal myocardium clearer. Our results showed that compared to that in the untargeted group, the normalized fluorescence intensity of cardiac damage detected by NIR in the targeted group increased 1.28-fold. The normalized fluorescence intensity increased 1.21-fold in the pretreatment group of the targeted groups. These data demonstrate the feasibility of applying pretreated fibrotic collagen and NIR contrast agents targeting TfR1 to identify ferroptosis at sites of cardiac injury, and its clinical value in the management of patients with MI needs further study.

## 1. Introduction

Acute myocardial infarction (MI) is a severe clinical emergency in which necrotic areas are replaced by fibrous tissue due to the limited regenerative capacity of cardiomyocytes, resulting in ventricular remodeling with severe effects on cardiac function [1]. Myocardial remodeling after MI is an important contributor to cardiac dysfunction, increased complications, and increased mortality, and myocardial fibrosis plays a key role in this process [2]. Therefore, early and effective prevention and treatment of myocardial fibrosis after MI are of great significance. Myocardial fibrosis refers to the excessive synthesis and deposition of extracellular matrix in the myocardium, which is mainly composed of type I and type III collagen. Oversynthesis of these collagens reduces myocardial compliance and consequently, cardiac function [3,4]. The occurrence of myocardial fibrosis after MI is not only related to the necrosis of cardiomyocytes, but also to the expression of a variety of cytokines and the activation of signaling pathways [4]. For example, hypoxia, inflammatory response, and oxidative stress may promote the occurrence of myocardial fibrosis. At present, the treatment methods for myocardial fibrosis after MI mainly include drug therapy, interventional therapy, and stem cell therapy. Drug therapy is commonly used, but the therapeutic effect is limited. Interventional therapy can improve myocardial perfusion, but it is not effective in the treatment of myocardial fibrosis [5]. Stem cell therapy is an emerging treatment method that promotes the regeneration and repair of cardiomyocytes by transplanting stem cells to improve cardiac function, which needs to be further verified [6]. At the same time, cardiac fibrosis may lead to changes in the density and blood flow distribution of the myocardial tissue. These changes may affect the distribution and metabolism of the imaging agent in the myocardium, thereby affecting the results of imaging. For example, in positron emission tomography/computed tomography (PET/CT), myocardial fibrosis may affect the uptake and clearance rates of the glucose metabolism of imaging agents in scar tissue, leading to inaccurate measurements of glucose metabolism rate [7]. Cardiac fibrosis may also affect the elasticity of myocardial tissue. Due to the different elastic coefficients, myocardial scar tissue and normal myocardial tissue will undergo different deformation when subjected to the same pressure or strain. This different deformation may lead to changes in the motion pattern of the myocardial tissue, which may affect the measurement results of techniques such as echocardiography [8]. 

Ferroptosis is not apoptosis or necrosis in the traditional sense, but a special mode of cell death. During this process, the normal balance of iron metabolism of cells is disrupted, leading to loss of cellular structure and function. It plays a crucial role in the development and progression of several cardiovascular diseases, such as MI, myocardial ischemia/reperfusion, and arrhythmia [9,10]. The cells facilitate the uptake of iron through transferrin (Tf), a protein that binds to iron, and its receptor 1 (TfR1). This process is initiated by the interaction between the two, leading to the endocytosis of complexes containing iron [11,12]. Changes in the expression of TfR1 and the disruption of iron transportation from the cytoplasm to mitochondria result in augmented mitochondrial iron levels, which are linked to ferroptosis subsequent to cardiac injury [13]. Therefore, TfR1 is expected to be a biomarker for the specific detection of MI-induced ferroptosis because it is associated with MI-induced damage. However, existing detection methods are insufficient to detect abnormal expression of TfR1 non-invasively and sensitively in cardiac injuries. New technical means are urgently needed to overcome this limitation. Molecular imaging can detect MI at the molecular and cellular levels and has become an important tool for diagnosing and treating MI. Advanced molecular imaging technology, including PET/CT [14] and magnetic resonance imaging (MRI) [15], has been designed to visualize changes in MI. Near-infrared (NIR) fluorescence molecular imaging has been utilized for visualizing cardiac injuries [16]. However, it is worth noting that the acquisition times for MRI and PET/CT were relatively long. Consequently, there is a demand to establish highly sensitive imaging techniques that can accurately detect ferroptosis in myocardial fibrosis. A TfR1-targeted optical contrast agent was then used to specifically “illuminate” the MI focus using NIR fluorescent molecular imaging, and experimental procedures have been detailed in Figure 1. Through our investigation, we found that the deposition and degradation of fibrotic collagen closely affected background signal interference in fibrotic tissue and uptake of TfR1-targeted contrast agents specifically designed to target MI within injured tissues. In the pathological process of myocardial fibrosis after MI, we first used human serum albumin-carried collagenase I (HSA-C) to degrade fibrotic collagen and reduce contrast agent retention in the cardiac fibrosis. The combination between HSA-C and TfR1-targeted peptides (CRT) fluorescence probe distinguished the sites of ferroptosis in myocardial injury accurately, and its normalized fluorescence intensity was 1.21-fold compared to that of CRT-ICG (CI) alone.

Myocardial fibrosis can significantly affect the identification of myocardial injury, and degradation of excessive fibrotic collagen can make the identification of myocardial injury more intuitive, so HSA-C is the key to solving this problem. At the same time, the development of functionalized molecular contrast agents with high sensitivity and specificity is expected to accurately identify cardiac damage after MI.

## 2. Materials and Methods

### 2.1. Synthesis and Characterization of HSA-C

The HSA-PEG-Maleimide (HSA-PEG-Mal) polymer was self-assembled using a method previously described in the literature [17]. To summarize, HSA-PEG-Mal was dissolved in dimethyl sulfoxide and gradually added to deionized water. The resulting solution underwent dialysis against deionized water for 24 h and was subsequently centrifuged. The supernatant was collected and stored at 4 °C. To thiolate collagenase I, the Traut reaction was employed by introducing 5 × Traut to the collagenase I solution (5 mg/mL in phosphate-buffered saline [PBS]) and stirring the mixture at 25 °C for 1 h. The obtained product, namely thiocollagenase I (SH-Collagenase I), underwent purification using a desalting column (Thermo Fisher Scientific, Waltham, MA, USA). Next, HSA-PEG-Mal was reacted with SH-collagenase I at 25 °C overnight to prepare pre-C-HSA. Then, Mal-PEG-COOH was added and reacted at room temperature for 4 h. SH-PEG-COOH was added to the solution and reacted at 25 °C for 4 h. The HSA-C solution was obtained and centrifuged for 30 min. The supernatant was removed, and the pelleted polymer was resuspended in PBS and stored at 4 °C. Using a Zetasizer Nano (Malvern Instruments, Malvern, UK) to characterize the hydrated particle size distribution and zeta potential.

### 2.2. Synthesis of CI

The Tfr1-binding peptide CRTIGPSVC (CRT, GL Biochem Co., Ltd., Shanghai, China) [18] underwent the labeling process by conjugating with indocyanine green-N-hydroxysuccinimide ester (ICG-NHS; Solarbio Co., Ltd., Beijing, China) at room temperature. Specifically, the separate dissolution of ICG-NHS and CRT in dimethyl sulfoxide (Solarbio Co., Ltd., Beijing, China) took place. The resulting mixture underwent overnight agitation in the absence of light, followed by a 24-h dialysis period to eliminate any unbound small molecule peptides CRT. Ultimately, the lyophilization process was conducted to acquire CRT-ICG.

### 2.3. Characterization of HSA-C and CI

To conduct detection, CI was dissolved in deionized water. The Zetasizer Nano ZS (Malvern, UK) was employed at 25 °C to measure the hydrodynamic diameters of HSA-C. For optical absorption analysis of the ICG-NHS (ICG-NHS was used as the CON) and CI, a UV–Vis–NIR spectrophotometer from Shimadzu (Kyoto, Japan) was utilized. To measure CON and CI’s emission and excitation spectra at 25 °C, a fluorescent spectrophotometer F-7000 from Hitachi (Tokyo, Japan) was employed. Fluorescence intensity analysis of CI at various concentrations was carried out using an IVIS spectrophotometer from PerkinElmer (Waltham, MA, USA).

### 2.4. Cell Counting Kit 8 (CCK8) Testing for Cytotoxicity

H9c2 cardiomyocytes were purchased from Wuhan Pronosay Life Technology Co., Ltd. (Wuhan, China), and all were species identification correct cells. H9c2 cells were cultured in a high-glucose medium (Dulbecco’s modified eagle medium, DMEM) containing 10% fetal bovine serum (FBS) (Gibco, Emeryville, CA, USA) and 1% penicillin–streptomycin (Sevier, Wuhan, China) at 37 °C in a 5% CO_2_ environment provided by the Heracell system from Thermo Fisher Scientific. Fluid exchanges were performed every 48 h. Cells within 10 passages were selected for experiments to ensure cell viability. When the cell density reached 90%, the cells were digested with 0.25% EDTA-containing trypsin, a complete medium was used to abort the cell digestion process, and the old culture medium was removed. The cells were resuspended in a complete culture medium, seeded in a 96-well plate at a density of 2 × 10^4^ cells/well, and incubated at 37 °C for 24 h. To investigate the effects of CI, the probes were diluted in a special medium above. The concentration of the probe varied from 0 to 80 μg/mL, increasing by 10 μg/mL each time. Both the CI group and the control medium group were included in the experiment. Following the 24 h incubation, the cells were washed thrice using PBS. Then, 10 μL of CCK8 was added and the cells were incubated in the special medium for an additional 2 h. Measuring the optical density (OD) values at 450 nm by a multimode microplate reader from BioTek, Winooski, VT, USA. The results were analyzed as a percentage of the control pore value.

### 2.5. Ethics

The animal studies in this research were performed in compliance with the guidelines outlined by the Institutional Animal Care and Use Committee of Xiyuan Hospital, China Academy of Chinese Medical Sciences, and ethical review approval (2021XLC027) was obtained.

### 2.6. MI Animal Model Establishment

In this study, we utilized 7-week-old male C57BL/6N mice (Beijing Vital River Laboratory Animal Technology Co., Ltd., Beijing, China). The animals were kept in SPF-grade animal houses at 20–25 °C and 40–70% relative humidity. Mice were housed in cages with 5 mice per cage. The animal house was kept quiet, and mice were fed and watered ad libitum, with a 12/12 h cycle of light. The model of MI was established by anesthetizing 60 mice with 2% isoflurane. To expose the myocardium, a precise incision was made on the left chest of the mouse between its third and fourth ribs, followed by careful separation of the muscles. Roughly 2 mm below the junction of the left atrial appendage and conus arteriosus, the ligation of the anterior descending branch of the coronary artery took place, and no ligation for the Sham group. After the surgery, the heart was returned to the heart cavity and the skin was stitched up. The MI mice were divided into three groups randomly after the electrocardiogram confirmed ischemia: (1) MI + CON, (2) MI + CI, and (3) MI + HSA-C + CI.

### 2.7. Western Blot

The RIPA protein extraction reagent was precooled and the protease inhibitor cocktail was added. The heart affected by the disease was sectioned into smaller pieces. The tissue was homogenized by 10% at 30,000 rpm for 2 min and lysed by shaking at 4 °C for 30 min. Centrifugation was performed at 13,000 rpm (4 °C) for 10 min. The resulting liquid was collected by centrifugation to obtain the supernatant. The supernatant was used for protein quantification and protein sample preparation. The concentration of protein in the supernatant was determined using the Pierce BCA Protein Assay Kit. To begin the electrophoresis process, a G protein solution of 20 µL was mixed with a solution of SDS (sodium dodecyl sulfate) loading buffer. A polyacrylamide gel with a composition of 4% to 15% was used for the separation of the protein molecules. Electrophoresis proceeded at 100 V for a designated amount of time ranging between 90 to 120 min. After the gel no longer showed any traces of the bromophenol blue dye, the electrophoresis halted, and the protein subsequently moved onto the PVDF (polyvinylidene difluoride) membrane. Treatment of the PVDF involved placing it in the solution sealed at 25 °C for 1 h, and allowing it to remain overnight at a temperature of 4 °C. This solution contained a diluted primary antibody that was used for specific protein detection (dilution ratio of 1:1000). A decolorization shaker was utilized to rinse the PVDF membrane three times with tris-buffered saline with tween-20 (TBST), with each rinse lasting 15 min at a temperature of 25 °C. To proceed further, the PVDF membrane underwent treatment with a secondary antibody that was diluted at a ratio of 1:5000. This treatment took place for a span of 1 to 2 h at a temperature of 25 °C. Subsequently, the membrane was subjected to three additional washes using TBST on a decolorization shaker, maintaining a temperature of 25 °C. Lastly, a chemiluminescence reaction detection process ensued. The resulting data were then meticulously recorded and subsequently analyzed utilizing Image J software (1.53k).

### 2.8. Masson Staining

Mouse myocardial tissues were washed three times with PBS and subsequently fixed with 4% paraformaldehyde solid solution. After complete fixation, the tissue was embedded using the paraffin embedding technique and then cut into 5 μm thick sections using a paraffin tissue microtome; paraffin sections were deparaffinized to the distilled water stage according to routine procedures. Weigert iron hematoxylin staining solution can be prepared by mixing reagents A1 and A2 (Solarbio Science & Technology Co., Ltd., Beijing, China) in a 1:1 ratio. This stain was dropped onto the sections to cover the sections and stained for 5–10 min. After completion of staining, sections were washed using distilled water to remove excess staining solution. Next, acidic ethanol differentiation solution was added dropwise for 5–15 s, followed by a 30-s wash with distilled water. Staining was performed using Ponceau-fuchsin staining solution for 5–10 min. In the staining process, the weak acid working solution should be prepared according to the ratio of distilled water: weak acid solution = 2:1. A weak acid working liquid drop was added to the section and the wash time was 30 s. After the excess liquid was poured off, phosphomolybdic acid solution was dropped for 1–2 min. It was then washed again with a weak acid working solution for 30 s. After the excess liquid was poured off, aniline blue staining solution was dropped and stained for 1–2 min. This was followed by another 30-s wash using a weak acid working solution. Finally, a rapid dehydration was performed using 95% ethanol for 2–3 s. This was followed by dehydration using absolute ethanol twice for 5–10 s each. The transparencies were then treated twice with xylene for 1–2 min each. Finally, neutral gum was used for sealing. Microscopy, image acquisition, and analysis were subsequently performed.

### 2.9. NIR Fluorescence Imaging

Before fluorescence imaging, the hair on the chest of the mice needs to be shaved, because the black mice themselves produce fluorescence self-absorption. The mice were anesthetized with a 2% isoflurane–air gas mixture for the entire imaging procedure. Mice injected with optical contrast medium were imaged using the IVIS spectrophotometer of PerkinElmer (Waltham, MA, USA) under the following conditions: Level-Hish, Em-840, Ex-745, Epi-illumination, Bin: (30) 8, FOV: 13.2, £2.0.63, Living Image Version: 4.7.4.21053 (14 August 2020), Camera: IS2105N8094, Andor, iKon. The acquired images can be used for noise removal and calculation of fluorescence signal intensity in the heart and liver using IVIS software (v4.4). The data were statistically analyzed by calculating the ratio of the fluorescence intensity of the heart to that of the liver, known as adaptive normalization.

### 2.10. Statistical Analysis

Mean ± standard deviation (SD) expresses the data. Paired *t*-tests were conducted within each group to make comparisons. One-way analysis of variance (ANOVA) was used for comparisons between groups, and multiple comparisons or *t*-tests were used as appropriate. *p* < 0.05 is a significant difference.

## 3. Results

### 3.1. HSA-C and CI’s Synthesis and Characterization

To verify the targeting of the CI probe, we performed a series of experiments. Firstly, we used western blotting (Figure 1A), and the results clearly showed that the expression level of TfR1 was significantly increased in the MI mouse model compared with the Sham group, which was statistically significant (Appendix A). This result demonstrates that the TfR1-targeting ligand can accurately recognize the target in the MI mouse model. To further verify this conclusion, we performed immunohistochemical staining (Appendix A) which showed that TfR1 was highly expressed at the site of myocardial infarction. We can intuitively observe that HSA-C has no obvious effect on tissues in MI mice, as seen in Appendix A. This result indicated that HSA-C did not cause significant tissue damage under the experimental conditions. Subsequently, we performed Masson staining experiments (Figure 1B). The staining results showed that there was a large amount of fibrous tissue in the MI model compared with the Sham group, indicating a higher degree of fibrosis. This is consistent with the experimental appearance and histopathological changes observed in the MI model, further validating our conjecture. We synthesized HSA-C that can degrade fibrosis, as well as CI, an optical probe targeting TfR1, and evaluated their biodistribution in vivo. The hydrated HSA-C particle sizes are shown in Figure 1C. After CRT peptide labeling, the UV-VIS absorption spectrum of CI changed and was enhanced at approximately 200 nm (Figure 1D). In addition, the binding of ICG to CRT did not alter its fluorescence properties; the fluorescence spectra of CON and CI are shown in Figure 1E. The NIR properties of different CI concentrations are of great significance in the detection of TfR1 in vivo. The NIR fluorescence spectral intensity (y = 1.57 × 10^6^x + 1.193 × 10^7^, R2 = 0.9817, *p* < 0.01) and the concentration of CI were linearly dependent (Figure 1F). This finding suggests that by monitoring the concentration changes of CI, we can indirectly understand the expression level of TfR1. In in vivo detection, this correlation has important implications for diagnosis and treatment monitoring of MI-induced cardiac injury. In addition, different concentrations of CI had no significant effect on H9c2 cell viability (Figure 1G).

### 3.2. NIR Fluorescence Imaging in the MI Mouse Model

In the mouse model of MI, significant differences were observed between the HSA-C-pretreated group and the untreated group. Masson staining showed that myocardial fibrosis was significantly reduced in the MI group after HSA-C pretreatment, indicating that HSA-C had a significant anti-fibrosis effect (Figure 2A). To further quantify this effect, we employed NIR fluorescence imaging for real-time monitoring. The data showed that after HSA-C pretreatment, the target probe at the cardiac site showed the highest fluorescence intensity at 48 h after CI injection into the tail vein. To explore this phenomenon in depth, we conducted a comparative analysis of the groups that received different treatments (Figure 2B and Appendix A). Compared with the MI + CON group, the normalized fluorescence intensity of the MI + CI group was significantly enhanced 1.28-fold. Compared with the untreated MI + CI group, the fluorescence intensity of the HSA-C pretreated MI + CI group was 1.21 times as much as that of the MI + CI group, which was statistically significant (Figure 2C). To evaluate the biodistribution of CI in vivo, we performed ex vivo fluorogram imaging, see Appendix A, and the results showed that CI signals were enriched in cardiac sites of the MI +HSA-C+CI. These data provide strong evidence that HSA-C pretreatment has a significant antifibrotic effect in the mouse model of MI and is effective in improving the normalized fluorescence intensity and distribution of the targeted probe CI at the site of the injured heart. This finding demonstrates the potential application of NIR fluorescence imaging technology in the evaluation of drug efficacy.

## 4. Discussion

Cardiac fibrosis is a common pathological process that negatively affects the structure and function of the heart. In the process of cardiac fibrosis, cardiomyocytes are gradually replaced by fibrous tissue which leads to a decrease in the systolic and diastolic function of the heart. In addition, cardiac fibrosis may also affect the sensitivity and specificity of molecular imaging. Due to the presence of fibrous tissue, some molecular imaging agents may not bind to cardiomyocytes normally, or the probes may be unspecifically deposited at the site of fibrosis, resulting in inaccurate imaging results. HSA-C can effectively improve cardiac fibrosis by inhibiting the synthesis of collagen, thereby reducing the formation of fibrous tissue. In the animal experiment, HSA-C could significantly reduce the degree of cardiac fibrosis. In addition, HSA-C can also improve the sensitivity and specificity of molecular imaging. Due to the reduction of fibrous tissue, injured cardiomyocytes can be better combined with specific probes to reduce nonspecific deposition of untargeted probes, thereby improving the accuracy of imaging results.

In recent years, ferroptosis, a new disease mechanism, has gradually become a research hotspot. This particular mode of cell death is not only closely related to common diseases such as cardiovascular diseases, tumors, and neurodegenerative diseases, but also involved in multiple other fields [9,19,20,21]. Several studies have demonstrated a close relationship between ferroptosis and cardiac ischemic injury. Due to the distinctive functions of excessive iron and oxidative damage to lipids in causing harm to the heart muscles, ferroptosis has emerged as both a potential etiology and a therapeutic strategy for myocardial injury associated with ischemic diseases [22]. Abnormal TfR1 expression and the ameliorative effect of iron homeostasis therapy on cardiac injury are associated with abnormal iron metabolism in multiple clinical trials [23,24,25]. The overexpression and alteration of iron transport play key roles in developing ferroptosis-related cardiac lesions. Therefore, we hypothesized that molecular imaging targeting TfR1 may be useful for accurately detecting MI injury. At present, there are few reports on molecular imaging in vivo detection methods for ferroptosis, which weakens the rationality and enthusiasm of developing drugs for the treatment of ferroptosis [26,27,28]. Therefore, the present study further proposes that TfR1 overexpression can be used as a biomarker of ferroptosis considering iron homeostasis in healthy participants. To explore biomarkers of ferroptosis more deeply, we introduced near-infrared fluorescence imaging. This technique has the advantages of high sensitivity and absence of ionizing radiation, which provide an ideal means for in vivo imaging. In this study, we designed a novel NIR fluorescence imaging strategy based on HSA-C degradation of myocardial fibrosis combined with ferroptosis detection. This strategy provides three important advantages for imaging: first, it reduces the uptake of nonspecific probes that injure cardiomyocytes and improves the sensitivity and specificity of the signal. Secondly, HSA-C can help to improve the condition of myocardial fibrosis, thus providing a new idea for treatment. In addition, we conducted experiments and data analysis to verify the validity and reliability of this strategy. By comparing with CI alone, we found that the HSA-C pretreatment + CI strategy increased the normalized signal intensity 1.21-fold, showing its superiority in improving the imaging effect.

Previous studies have described methods for visualizing ferroptosis using PET/CT or MRI [29,30,31]. To our knowledge, this is the first study to use HSA-C and CI to prepare a green, safe, material-specific, and simple method for synthesizing myocardial injury imaging probes in vivo. The HSA-C pretreated CI probe strategy can detect abnormalities in intracellular TfR1 levels following MI-induced cardiac injuries. This method offers a robust approach to circumvent the impact of cardiac remodeling and assess cardiac lesions. It enables real-time evaluation of cardiac injuries following MI, facilitating timely adjustment of treatment plans and enhancing therapeutic outcomes. Although we have made some progress in the study of myocardial injury in MI, there are still some limitations and challenges. First, our study has so far only been validated in a mouse model, which warrants further validation in large animal models and humans. This is because although the mouse model can provide us with a certain reference, there are significant differences from humans in physiology and metabolism. Therefore, more extensive and in-depth studies are needed to ensure the validity of our findings in humans. The HSA-C pretreatment CI probe strategy that we currently use may not apply to all models of myocardial injury. This is because different types of myocardial injury may have different mechanisms and processes, so appropriate methods and techniques need to be selected for specific situations. To address this issue, we need to further optimize and improve the existing probe strategy so that it can be better applied to various models of myocardial injury. In addition, further monitoring is also needed to target the possible adverse effects of HSA-C on cardiac injury after MI.

In summary, a novel HSA-C and CI imaging strategy was established to assess MI-induced myocardial injury in vivo. This study has important theoretical significance and practical value, and provides new ideas and methods for the research and treatment of in vivo visualization of myocardial ferroptosis. We will continue to study the mechanism of myocardial injury, in order to bring better detection methods for patients with myocardial injury and improve the treatment effect.

## Data Availability

After careful consideration, we have added research data to the additional information to ensure the completeness and accuracy of the information.

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
