# Peer review of "Enhanced TfR1 Recognition of Myocardial Injury after Acute Myocardial Infarction with Cardiac Fibrosis via Pre-Degrading Excess Fibrotic Collagen"

_biology, 2024, doi:10.3390/biology13040213_

Round 1
Reviewer 1 Report
Comments and Suggestions for Authors
In this manuscript, the authors developed an imaging strategy to mark the damaged heart area after acute MI. They pretreated myocardial fibrotic collagen with collagenase I combined with human serum albumin (HSA-C) and subsequently visualized the site of cardiac injury by near-infrared (NIR) fluorescence imaging using an optical contrast agent (CI, CRT-indocyanine green) targeting transferrin receptor 1 peptides (CRT). The pretreatment with HSA-C can reduce background signal interference in the fibrotic tissue while enhancing CI uptake at the heart lesion site, making the boundary between the injured heart tissue and the normal myocardium clearer. Their data showed a 1.28-fold increase in the normalized fluorescence intensity of cardiac damage detected by NIR in the targeted group compared with that in the untargeted group. The normalized fluorescence intensity increased by 1.21-fold in the pretreatment group of the targeted groups. Last, the authors concluded that their data suggests the feasibility of applying pretreated fibrotic collagen and NIR contrast agents targeting TfR1 to identify ferroptosis at sites of cardiac injury and its clinical value in the management of patients with MI needs further study.
Altogether this is a straightforward study. However, the manuscript was poorly written and lacked important information when describing/interoperating their experiments. My comments are listed as below.
1) The first concern is the usage of HSA-collagenase I to clear the fibrosis. Since the Scheme 1 shows that HSA-C will be injected to the animals before the following labeling and imaging, the extra collagenase I in the circulatory system will non-specifically digest collagen which is a critical component of extra cellular matrix of many tissues, which will lead to potential tissue damage.
2) In Fig.1A, a graph showing the quantification of TfR1 expression in MI and Sham groups needs to be shown in order to conclude whether the difference was significant.
In addition, was the western blot performed using the entire heart tissue or just the tissue at the injured site? The increased TfR1 expression could resulted from the surrounding uninjured heart tissue. A specific staining for TfR1 on the tissue sections of the MI hearts will clearly reveal if TfR1 was indeed upregulated at the injured site.
3) The authors stated that they evaluated the biodistribution of CI in vivo. How this in vivo evaluation was conducted was not clearly described. In addition, representative heart tissue images for Fig.1D should be shown to support that the CI signal accumulated at the injured site.
4) In Fig.2A, the authors stated that HSA-C had a significant anti-fibrosis effect. However, the MI and MI+HSA-C heart seems way less injured than that in MI heart based on the images shown. In addition, the zoomed-in images were taken by bias. The box in MI was in the center of the fibrosis tissue whereas the box in MI+HSA-C was at the junction between the healthy and fibrosis tissue. The images should be taken in the hearts with similar level of injury and taken at the similar place within the fibrosis tissue for both groups.
5) In Fig.2B and 2C, how the fluorescence intensity was normalized is not shown. Furthermore, despite the difference was significant, the signal intensity was improved by only ~ 28% or 21% which are mild change, questioning the value of the imaging protocol proposed in this study.
Comments on the Quality of English LanguageThe manuscript was poorly written and lacked important information when describing/interoperating their experiments. The writing needs to be improved to reach the publication standards.
Minor comments:
1) There is a typing error in 3rd row of the abstract. I assume the authors were stating marking (instead of making) clear heart damage from the fibrosis micro-environment.
2) In the figure legends of Scheme 1 and Fig.2, the text fonts are not consistent.
Reviewer 2 Report
Comments and Suggestions for Authors
In this experimental in cell culture and animal model systems report
the authors proposed the pretreatment of myocardial fibrotic collagen
with collagenase I and human serum albumin (HSA-C) to reduce background
signal interference in near-infrared (NIR) fluorescence imaging. The
authors assessed the iron up-taker transferrin receptor 1 (TfR1) which
is related to ferroptosis when the mitochondria iron levels are increased.
Concerns
1. The results of Figure 1 and Figure 2 collectively suggest an
improvement of near-infrared (NIR) fluorescence imaging intensity,
but the authors should addresse the cost-effectiveness of their
proposed protocol for the use of HSA-C. Is the improvement
of NIR fluorescence imaging intensity enough to justify the
preparation or purchase of HSA-C for this process?
Reviewer 3 Report
Comments and Suggestions for Authors
In the submitted manuscript, titled "Enhanced TfR1 recognition of myocardial injury after acute myocardial infarction with cardiac fibrosis via pre-degrading excess fibrotic collagen", authors described an imaging strategy by pretreating myocardial fibrotic collagen with collagenase I combined with human serum albumin and subsequently visualized the site of cardiac injury by near-infrared (NIR) fluorescence imaging using an optical contrast agent targeting transferrin receptor 1 peptides. There are some issues needed to be addressed as following,
1. In section 2.2, authors should explain that CRT without integrating with ICG-NHS can be removed by dialysis?
2. In Fig. 2A, the scale bar is missing.
3. Fig.2, the Sham+CON data should been added.
4. In Fig.2, the fluorescence intensity of the MI + CON group was enhanced compared with the Sham + CI group, and it should be explained why ICG was enriched at the site of myocardial injury.
5. whether the increase of fluorescence intensity in MI + CI group compared with MI + CON group was due to the up-regulation of CRT caused by HAS-C pretreatment? Additional experiments should be performed to show that HAS-C pretreatment does not lead to CRT upregulation.
Comments on the Quality of English LanguageMinor editing of English language required
Round 2
Reviewer 1 Report
Comments and Suggestions for Authors
In this manuscript, the authors developed an imaging strategy to mark the damaged heart area after acute MI. They pretreated myocardial fibrotic collagen with collagenase I combined with human serum albumin (HSA-C) and subsequently visualized the site of cardiac injury by near-infrared (NIR) fluorescence imaging using an optical contrast agent (CI, CRT-indocyanine green) targeting transferrin receptor 1 peptides (CRT). The pretreatment with HSA-C can reduce background signal interference in the fibrotic tissue while enhancing CI uptake at the heart lesion site, making the boundary between the injured heart tissue and the normal myocardium clearer. Their data showed a 1.28-fold increase in the normalized fluorescence intensity of cardiac damage detected by NIR in the targeted group compared with that in the untargeted group. The normalized fluorescence intensity increased by 1.21-fold in the pretreatment group of the targeted groups. Last, the authors concluded that their data suggests the feasibility of applying pretreated fibrotic collagen and NIR contrast agents targeting TfR1 to identify ferroptosis at sites of cardiac injury and its clinical value in the management of patients with MI needs further study.
After revision, the authors had substantially revised the manuscript and addressed my concerns. I have a few minor comments listed as below.
1) There are still few type errors such as lines 96 and 166.
2) In line 103-104, the authors should cite the references or figures in this manuscript to support that TfR1 is increased in the infarcted region.
Comments on the Quality of English LanguageThere are few typing or spelling errors needed to be carefully corrected to reach the publication standards.
Reviewer 2 Report
Comments and Suggestions for Authors
In this experimental cell culture and animal model systems report
the authors proposed the pretreatment of myocardial fibrotic collagen
with collagenase I and human serum albumin (HSA-C) to reduce background
signal interference in near-infrared (NIR) fluorescence imaging. The
improvement is significant and facilitates observation.
Author Response
We sincerely thank the reviewer for your affirmation and valuable comments. Under your professional guidance, we conduct in-depth research and fine polishing, so that the quality of the manuscript can be significantly improved. Thank you again for your contribution.